# A Single-Fed Wideband Circularly Polarized Cross-Fed Cavity-Less Magneto-Electric Dipole Antenna

**DOI:** 10.3390/s23031067

**Published:** 2023-01-17

**Authors:** Linyu Cai, Kin-Fai Tong

**Affiliations:** Department of Electronic and Electrical Engineering, University College London, Torrington Place, London WC1E 7JE, UK

**Keywords:** circular polarization, wideband, stable high gain, magneto-electric dipoles, cross-fed

## Abstract

In this paper, we proposed a new wideband circularly polarized cross-fed magneto-electric dipole antenna. Different from conventional cross-dipole or magneto-electric dipole antennas, the proposed simple geometry realizes a pair of complementary magnetic dipole modes by utilizing the two open slots formed between the four cross-fed microstrip patches for achieving circular polarization and high stable gain across a wide frequency band. No parasitic elements are required for extending the bandwidths; therefore, both the radiation patterns and in-band gain are stable. The simulated field distributions demonstrated the phase complementarity of the two pairs of magnetic and electric dipole modes. A parametric study was also performed to demonstrate the radiation mechanism between the electric and magnetic dipole modes. The radiating elements are realized on a piece of double-sided dielectric substrate fed and mechanically supported by a low-cost commercial semirigid cable. The overall thickness of the antenna is about 0.22λ_o_ at the center frequency of axial ratio bandwidth. The measured results show a wide impedance bandwidth (|*S*_11_| < −10 dB) of 70.2% from 2.45 to 5.10 GHz. The in-band 3-dB axial ratio bandwidth is 51.5% from 3.0 to 5.08 GHz. More importantly, the gain of the antenna is 9.25 ± 0.56 dBic across the 3-dB axial ratio bandwidth.

## 1. Introduction

Antennas with wide bandwidths and stable high gain features are key devices for achieving the high data rate and low latency in future wireless communications and sensing systems [1,2]. These features can compensate the free-space path loss and improve the signal stability and reliability for ensuring the quality of systems. Magneto-electric (ME) dipole antennas, which have a pair of slightly separated resonances between their magnetic dipole (M-dipole) and electric dipole (E-dipole) modes, are well-known for providing such advantages [3,4,5]. In addition to the gain improvement contributed by the reflection from the ground plane, the complementary radiation of the M- and E-dipole modes of the antennas can further increase the gain and maintain stable radiation patterns across the wide passband [6,7]. Various wideband linearly polarized (LP) ME dipole antennas have been reported [8,9,10]; however, circularly polarized (CP) designs are rare.

In a multipath-rich environment, the polarization of the signal will be altered at reflections. Polarization loss will happen when the polarization of the signal and antennas are not aligned. CP antennas have been widely used to solve such problems [11,12]. As no polarization alignment is required, the wireless channels are more stable and robust. However, for single-fed CP antennas, the axial ratio (AR) bandwidth is usually narrow, typically less than 10% [13,14]. Wideband CP ME dipole antennas have recently been drawing attention [15,16,17,18,19]. An aperture coupled CP ME dipole antenna array fed by a low-loss gap waveguide has achieved 14.5% of AR bandwidth [16]. A dielectric-based ME dipole antenna achieves a wide impedance bandwidth of 56.7% (VSWR ≤ 2) and 3-dB AR bandwidth of 41%. However, the gain fluctuation is about ±2.6 dBic within the operating frequency band [20]. A dual-fed bowtie dipole utilizing a wideband 90° hybrid Wilkinson power divider feed network was proposed in [21]. By introducing two metallic strips connecting the bowtie dipoles, the gain is improved to 9 dBic with the help of a relatively large ground plane. However, wideband CP ME dipole antennas with stable (less than ±1 dB variation) and high gain (larger than 9 dBic) with reasonable footprint (about one square wavelength) are rare.

Recently, wideband crossed-dipole CP antennas with parasitic elements have also been proposed for improving the 3-dB AR bandwidth [22,23,24,25]. In [22], a similar geometry of four cross-fed rectangular patches was reported. As the open slots are not aligned in a straight line, the cross-dipole design solely relies on the addition of curved-delay line to provide the required phase difference in order to enhance the AR bandwidth. Large ground plane, cavity reflectors, and the addition of parasitic elements are also utilized to boost the gain and bandwidths. However, without the presence of the complementary M-dipole modes as proposed, these designs either suffer from low gain and/or large gain fluctuation within the passband. Some crossed-dipole antennas utilize complex cavity reflectors to mitigate the fluctuating gain problems [26,27], but the improvement is limited, and the antenna geometries become complicated. A comparison of the measured results of recently reported wideband CP antennas and the proposed antenna is presented in Table 1. It can be observed that the proposed antenna demonstrates wide impedance and 3-dB AR bandwidths without suffering from low and/or high gain fluctuation within the bandwidth. The simple single-fed geometry can also reduce the fabrication cost and potential errors, and additionally, no cavity reflector is required for achieving the stable high gain.

On the other front, as described in [5], the M-dipole mode of a general ME dipole antenna is usually realized by a pair of vertical cavity walls. A linearly polarized ME dipole antenna without the conventional quarter-wavelength vertical wall was reported [34]; it demonstrated that the gap between the wide E-dipole patches, operating as an open slot, can replace the quarter-wavelength cavity walls for generating the M-dipole mode. More importantly, the wide bandwidth and stable high gain features can be retained without the vertical walls. The profile of the cavity-less ME dipole antenna can also be reduced to 0.16 λ_o_ at the center frequency. With the facilitation of a pair of cross-dipoles, the cavity-less concept in [34] is extended to circular polarization in this paper. The proposed wideband CP cavity-less ME dipole antenna is designed to operate at 5G mid-band. CST Studio Suite 2020 [35] was used to perform the simulations.

In Section 2, the antenna geometry and operating principle will be presented. The parametric study of the critical parameters of the antenna in Section 3 will confirm the operating principle and serve as a reference for designing the proposed antennas for different applications. The simulated and measured results of the final design are provided in Section 4. The work is concluded in Section 5.

## 2. Geometry and Operating Principle

### 2.1. Antenna Geometry

Figure 1 shows the isometric view of the proposed antenna; the detailed dimensions are provided under the caption. The proposed antenna consists of three components: (i) The radiating layer accommodating Patches A, B, C, and D is a piece of Rogers RT5880LZ substrate (*ε_r_* = 1.96, *h_sub_* = 1.27 mm and tan *δ* = 0.0019@10 GHz). Patches A and B are on the top side, whereas Patches C and D are on the bottom side of the radiating layer. (ii) A semirigid coaxial cable (RG402, 50 Ω, ∅ 3.58 mm), which is connected to the corresponding patches through the two crossed-feeding networks. (iii) A square aluminium planar ground plane is located about a quarter-wavelength under the radiating layer. The 1 mm thick ground plane also serves as a reflector of the backward radiation; in addition to the complimentary of electric and magnetic dipole modes, it provides a unidirectional radiation pattern and higher front-to-back ratio.

Figure 1 also shows the detailed metallic patterns of the radiating elements and crossed-feeding network on the top and bottom sides of the radiating layer of the realized prototype. Patches A, B, C, and D are basically four square patches and have the same dimensions with length *L_patch_*. Through the cross-feeding network, Patches A and B are connected to the inner conductor of the semirigid coaxial cable, and Patches C and D are connected to the outer conductor of the cable and the ground plane of the antenna. Avoiding the short-circuit to the center conductor, a hole on the short lead of the bottom cross-feeding network is etched.

The four patches are designed to operate as half-wave electric printed dipoles; therefore, the length of patch (*L_patch_*) was first assigned as roughly a quarter-wavelength at the center frequency in the simulations. Extending the concept reported in [27], which showed that the gap between the wide electric dipole can replace the vertical walls for creating the magnetic dipole, the two gaps between the four patches will form a pair of crossed magnetic dipoles, in the form of an open slot, for generating the circular polarization in this configuration. Together with the two crossed electric dipoles formed by the four patches and proper phase arrangement, four resonances contributing to the wideband CP radiation can be generated. The crossed-feeding network, which has two short and two long leads, is located at the center of the radiating layer. It plays a critical role in distributing equal power and proper phase to each patch to achieve circular polarization. The length of long leads is roughly a quarter wavelength. Such arrangement will introduce a 90° phase difference between Patches A and B and Patches C and D; therefore, it always keeps a 90° phase difference between adjacent patches for CP radiation. The length of the long leads is determined by its radius *R_ring_*. The widths of the long lead *W_ring_* and the short lead *W_con_* are adjusted for best impedance matching.

Finally, the square ground plane is set at about a quarter-wavelength below the radiating layer, so that the gain will be increased. Furthermore, the size of the ground plane is set at about one wavelength to further improve the gain of the ME dipole antenna. The finalized values of the parameters will be further discussed in Section 3.

### 2.2. Operating Principle

Conventionally, circular polarization is generated by two orthogonal resonances; they should have equal magnitude and a phase difference of 90°. The two resonances operate at slightly different frequency points for wider 3-dB AR bandwidth. In contrast, the proposed antenna has two pairs of 45° angularly separated modes, as shown in Figure 2a. The two electric dipoles operate at a lower frequency band as a result of their longer effective electrical length, whereas the two magnetic dipoles, in the form of open-end slots between the patches, operate at the higher frequency band, as shown in Figure 2a; therefore, two minimum AR frequency points and wide AR bandwidth were achieved, as shown in the AR plots.

In the proposed wideband CP cavity-less ME dipole antenna, the two pairs of rhombic-shaped electric dipoles are located on the diagonals of the square radiating layer. Patches A and C are connected to the short leads of the crossed-feeding network, as shown in Figure 1. On the other diagonal lie Patches B and D; they are the two arms of the Electric Dipole 2 and are fed by the two long leads. As the length of the long leads is roughly a quarter-wavelength longer than the short leads, the phase of signal traveling to Electric Dipole 2 is 90° delayed when compared to that of Electric Dipole 1. Through such an arrangement, CP radiation in the lower frequency band could be achieved.

The two open-end slots, which are formed by the gaps between the patches (highlighted in light blue in Figure 2a), radiate as magnetic dipole antennas. As shown in Figure 2a, Magnetic Dipole 1 is located along the *x*-axis, whereas Magnetic Dipole 2 is in the *y*-direction. Through the same crossed-feeding network, a 90° phase difference can be achieved between the two magnetic dipole modes. In considering the shorter length of the open slots, CP radiation at the high-frequency band is generated. The phase relations between the electric dipoles and magnetic dipoles were also illustrated in Figure 2a by assuming the phase of Patch A is 0°, which explains how the electric field can be produced in the open-end slots. The simulated *E*-field at a different time of the period (*T*) shown in Figure 2b verified the above explanations.

To further illustrate the four modes contributing to the wide impedance and 3-dB AR bandwidth of the proposed antenna, Figure 3 shows the simulated real and imaginary parts of the impedance of the proposed antenna. It can be observed that the four modes operate at around 2.8 GHz, 3.6 GHz, 4.4 GHz, and 4.9 GHz, respectively. As the two electric dipoles are operating at the diagonal direction on the *xy*-plane, in addition to the fringing field that exists between the edges of the electric dipoles and the ground plane, their effective lengths are longer than that of the magnetic dipoles. Therefore, the two electric dipoles operate at lower frequency (2.8 GHz and 3.6 GHz), and magnetic dipoles operate at the high-frequency band (4.4 GHz and 4.9 GHz). Depending on the frequency separation between the modes, the imaginary part of the 2nd E-dipole and 2nd M-dipole modes are only close to, but not equal to, zero. Further verifications will be provided in the Parametric Study in Section 3. By combining the resonances with the proper physical parameters, wide impedance bandwidth and 3-dB AR bandwidth can be achieved.

Figure 4 illustrates the electric fields at a different time (*t*) in one period of oscillation (*T*) of the proposed wideband CP antenna at the center frequency point of the 3-dB AR bandwidth, i.e., 4 GHz. It can be observed that the E-field rotates in an anticlockwise direction, with the wave propagating in the positive *z*-direction, i.e., a right-hand circular polarization. Left-hand circular polarization could be achieved by mirroring the radiating layer against the *y*-axis.

## 3. Parametric Study

A parametric study is presented in this section to further demonstrate the operating principle of the proposed wideband CP ME dipole antenna. Firstly, the four patches on the radiating layer play a key role in generating the fundamental resonances which provide the stable high gain across the wide operating frequency band, so the length of patch (*L_patch_*) is studied first. The crossed-feeding network, enabling the circular polarization and wide 3-dB AR bandwidth, is essential for wideband impedance matching; the width of the short leads (*W_con_*), width of the long leads (*W_ring_*), and radius of the arc (*R_ring_*) are then investigated. Thirdly, the height between radiating layer and the ground plane (*h_air_*) and the length of square ground plane (*L_ground_*) were also investigated to study their influence on gain and bandwidths.

### 3.1. Length of the Patches (L_patch_)

The four patches on the radiating layer are the main radiating elements. Together with the two narrow gaps between the patches, four resonant modes are produced, so the size of patches determines the frequencies of four resonances. Figure 5a depicts the real and imaginary parts of the impedance with different *L_patch_*. Since the resonant frequency of the half-wavelength electric dipole and the magnetic dipole are determined by this parameter, when *L_patch_* is longer, all four modes move to a lower frequency band. Although the resonant frequency changes with *L_patch_* varying from 15.5 mm to 20 mm (19.5 λ_g_ to 25.2 λ_g_), the value of impedance is generally not affected significantly, except for in the fourth mode, so the bandwidth is still wide generally, while the high-frequency band is affected more, as shown in Figure 5b,c. By tuning *L_patch_*, the operating frequency of the proposed antenna shifts accordingly, but the impedance bandwidth remains wide at about 60%.

### 3.2. Width of the Short Lead (W_con_)

The cross-feeding network is the critical part that transfers the properly split signal from the semirigid coaxial cable to each patch and the open-end slots. It plays an essential role in creating the 90° phase difference and allocates equal power to the four patches to generate circular polarization. The signal is spilt into four paths and travels to each patch through the feed network. Thus, the width of each lead becomes a critical value that affects the impedance matching.

Figure 6 shows the influence of *W_con_* on Z_11_ and S_11_. Generally, the wider *W_con_* is, the wider the S_11_ bandwidth will be. As can be observed from Figure 6a, larger values of *W_con_* will cause both the real and the imaginary part of the impedance of the antenna that is balanced and close to 50 + j0 Ω. In this design, 5.2 mm (0.065 λ_g_) is almost the maximum width that can be achieved, as it is limited by the space available. Figure 6c shows that by tuning *W_con_* from 3.7 mm (0.046 λ_g_) to 5.2 mm (0.066 λ_g_), the gain and AR are not significantly affected, although the gain at the high-frequency band slightly increased. The AR remains low in the investigated range of *W_con_*, and the AR profile shows that *W_con_* is related to the balance of AR between the low-frequency and high-frequency band. To sum up the analysis state so far, keep the following in mind when setting up the value of short lead *W_con_*: Firstly, the width should be as wide as possible if a wide impedance bandwidth is desired. Secondly, based on the required AR bandwidth, choose the value of *W_con_* for the antenna for a balanced circular polarization between the low- and high-frequency bands.

### 3.3. Width of the Long Lead (W_ring_)

Figure 7a,b show the influence of *W_ring_* on Z_11_ and S_11_, respectively. *W_ring_* affects the real part of the second and fourth modes and the imaginary part of the first and the third modes of the antenna. Generally, a wider *W_ring_* means wider impedance bandwidth in the investigated range, and 1.5 mm is almost the maximum width that can be achieved in this design.

In Figure 7c, *W_ring_* is tuned from 0.6 mm (0.008 λ_g_) to 1.5 mm (0.019 λ_g_); the gain remains high except in the high frequency since the impedance matching worsens when *W_ring_* is narrow. Because *W_ring_* affects the second and fourth modes of the proposed antenna, both circular polarizations could be affected. Relatively narrow *W_ring_* separates, while wide *W_ring_* brings the two AR center points closer.

When deciding the values of *W_con_* and *W_ring_*, it is always good to keep a large value so that the antenna will have a wide impedance bandwidth. More importantly, the *W_con_* and *W_ring_* affect the AR more significantly than the impedance bandwidth. The final optimized value should be carefully selected according to the AR, as wide AR bandwidth is the priority of this wideband CP cavity-less ME dipole antenna.

### 3.4. Radius of the Quarter-Wavelength Arc (R_ring_)

*R_ring_* represents the radius of the quarter-wavelength arc from center to the middle of the arc. The signal from the feed network travels along a longer path to Patches B and D due to the extra length of the long lead. Thus, the length of the long lead is the key for creating the 90° phase difference. Figure 8a,b show the influence of *R_ring_* on *Z*_11_ and *S*_11,_ respectively. It can be observed that the changes of *R_ring_* affect all four modes of the antenna. Shorter *R_ring_* makes the impedance match better at the high-frequency band, and wide bandwidth will be achieved.

Figure 8c shows the gain and axial ratio of the antenna when *R_ring_* varies from 4.9 mm to 5.5 mm. The gain remains high, and the gain in the high frequency decreased slightly since the impedance does not match well. *R_ring_* determines the phase difference between the orthogonal resonances, so the two circularly polarized frequency points could be affected. A shorter *R_ring_* brings the two AR center frequency points closer and results in a narrower 3-dB AR bandwidth. Simultaneously, a desirable value of *R_ring_* will arrange the two AR center points to be properly separated so that a wide AR bandwidth can be achieved.

### 3.5. Radiating Layer to Ground Plane Separation (h_air_)

*h_air_* represents the height between the radiating layer to the ground plane. Figure 9 shows the influence of *h_air_* on *Z*_11_ and *S*_11_ of the proposed antenna. By tuning *h_air_* from 14 mm (0.177 λ_g_) to 23 mm (0.290 λ_g_), the two lower electric dipole modes are affected more, as the fringing field at the edges is reduced at shorter *h_air_*, while the two magnetic dipole modes are less subjected by the change of *h_air_*. Therefore, the lower 3-dB AR bandwidth decreased with *h_air_*. In contrast, since the value of *Z*_11_ remains relatively stable and close to 50 + *j*0 Ω, the impedance bandwidth remains wide.

### 3.6. Length of Ground Plane (L_ground_)

Figure 10 depicts the influence of *L_ground_* on the proposed antenna. A square ground plane with a size equal or larger size than $\lambda_{0}^{2}$ will usually give better gain enhancement [6]. The value of *L_ground_* was set from 40 mm (0.504 λ_g_), i.e., same size of the radiating layer, to 100 mm (1.25 λ_0_) for investigation. Figure 10a,b show that Z_11_ and S_11_ are barely affected by the length of the square ground, except in the case of 40 mm since it is only about a half-wavelength of the low-frequency band. Figure 10c shows the gain and AR with different values of *L_ground_*. The 3-dB AR in the lower band is slightly improved when *L_ground_* varies from 60 mm to 100 mm, as more fringing field can reach the ground plane. However, the gain of the antenna is affected more by the size of the ground plane size. In the investigation range, the gain increased with a larger size of the ground plane. When the size increased to 100 mm, the peak gain was about 9.9 dBic at 4 GHz. It is worth mentioning that when the size of the ground plane is extended to 1.3 λ_0_, a slightly wider simulated 3-dB AR bandwidth and high peak gain can be achieved. In this paper, *L_ground_* = 80 mm was selected to strike the right balance between the size and performance.

## 4. Measurement Results and Discussions

Antenna prototypes, whose physical parameters are referenced to the results of the parametric study, were fabricated and measured to verify the performance of the proposed antenna. Figure 1 shows one of the fabricated CP cavity-less ME dipole antenna prototypes mounted inside the anechoic chamber during the measurement. Three prototypes were fabricated for the measurements, and the average values were taken to improve the accuracy.

The simulated and measured *S*_11_ results present a reasonable consistency, as shown in Figure 11. The measured impedance bandwidth (|*S*_11_| < −10 dB) is 70.2% from 2.45 to 5.10 GHz. Figure 12 shows the gain and 3-dB AR of the proposed antenna; the gains of the three antenna prototypes were measured by using the well-known three-antenna method, i.e., eliminating the duplicated transmitted power and antenna separation in the Friis equation in three sets of measurements. It can be observed that the measured average gain is about 9.31 dBic, with stable gain ±0.56 dBic across the 3-dB AR bandwidth. The maximum measured gain of the antenna is about 9.9 dBic at 4.2 GHz. The axial ratio of the three antenna prototypes were measured by rotating the wideband linearly polarized dual-ridge horn antenna to different axial angles. The measured 3-dB AR bandwidth is 51.5% from 3.0 to 5.08 GHz, which is wider than the simulated results. In addition to the discrepancy in fabrication, the difference may also be caused by the misalignment when rotating the antenna from *ϕ* = 0° to 45°, 90°, and 135° for AR measurement and the averaging effect between the results of the three prototypes built. However, the trend and shape of the results are reasonably close to each other.

To ensure of the stability of a wideband communication system, the radiation pattern of a wideband antenna at different frequency points within the operating bandwidth is expected to be consistent across the bandwidth. Figure 13 shows the simulated and measured radiation patterns at low- (3.2 GHz), mid- (4.0 GHz), and high- (4.8 GHz) frequency points of the 3-dB AR bandwidth. The co- and cross-polarization radiation patterns were measured at the four major planes, i.e., *ϕ* = 0°, 45°, 90°, and 135°. The measured results have the acceptable consistency with the simulated results. From Figure 13, high symmetry can be observed, the front-to-back ratio is about 20 dB in different planes over the passband, and the cross-polarization is also acceptable in general.

Table 2 compares the simulated and measured results of 3-dB beamwidth at the low-, mid-, and high-frequency points. Two parameters, |∆_1_| and |∆_2_|, were introduced for comparing the radiation patterns at different frequency points and principal planes. |∆_1_| represents the 3-dB beamwidth variations across the frequency in the same plane, whereas |∆_2_| is the variations in different planes at the same frequency. |∆_1_| shows good consistency at each plane, with a small variation between 9.2° to 12.1°, whereas the angular variation at different planes |∆_2_| ranges from 1.8° to 4.3°. This means the proposed CP cavity-less ME dipole antenna has a similar 3-dB beamwidth regarding the four principal planes over the frequency band. As explained, the low-frequency band mainly is contributed by the E-dipoles, while the high-frequency band is the result of the M-dipoles. Therefore, circular polarization should have the most average distribution at the center frequency (around 4 GHz), and that explains that |∆_2_| has the minimum value of 1.8° at the center frequency.

It is worth mentioning that usually the 3-dB beamwidth of a wideband antenna decreases at the higher frequency band due to the electrically larger radiation aperture when frequency increases. However, in this wideband CP cavity-less ME dipole antenna, the 3-dB beamwidth increases with frequency. The reason explaining this feature is that the M-dipole modes operate in the high-frequency band, and the aperture for the M-dipole is determined by the parameter 2 × *L_patch_* + *W_gap_*, not the diagonal of the radiating layer. Therefore, the aperture of the antenna responsibility to the radiation at the high-frequency band is not electrically increased accordingly, so the 3-dB beamwidth at the high-frequency band does not decrease.

## 5. Conclusions

In this paper, a novel wideband circularly polarized cavity-less magneto-electric dipole antenna with stable high gain was presented. By utilizing the four square-patches and the corresponding gaps between the patches to realize the two pairs of properly power-fed and phase-separated electric dipole and magnetic dipole modes, the proposed antenna has a wide impedance bandwidth of 70.2% from 2.45 GHz to 5.1 GHz. The measured AR bandwidth achieved is 51.5% from 3.0 GHz to 5.1 GHz, and its height equals about 0.24 λ_o_ at the center frequency of 3-dB axial ratio bandwidth. Inheriting the advantages from the cavity-less ME dipole antenna concept, the average gain is 9.31 dBic, and the in-band variation is only ±0.56 dBic. The proposed antenna has a simple geometry, which has no parasitic elements or cavity ground plane, for low-cost and easy fabrication.

## Figures and Tables

**Figure 1 sensors-23-01067-f001:**
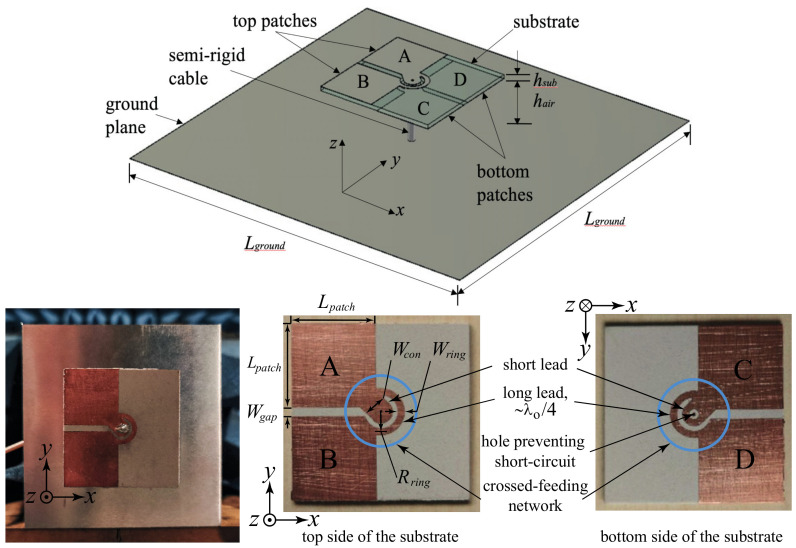
Geometry and fabricated prototype of the wideband CP cavity-less ME dipole antenna. *L_ground_* = 80.0, *h_sub_* = 1.27, *h_air_* = 17.7 (0.24 λ_0_), *L_patch_* = 18.5, *W_gap_* = 2.1, *R_ring_* = 5.2, *W_ring_* = 1.5, *W_con_* = 5.2. All dimensions are in millimeters.

**Figure 2 sensors-23-01067-f002:**
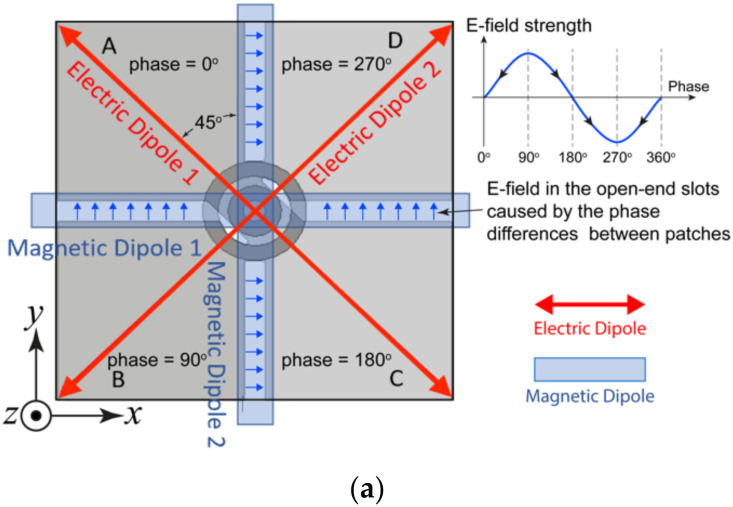
(**a**) The four fundamental modes of the proposed CP cavity-less ME dipole antenna, (**b**) *E*-field of the magnetic dipole modes at *t* = 0, *T*/4, *T*/2, and *t* = 3*T*/4 at 4 GHz.

**Figure 3 sensors-23-01067-f003:**
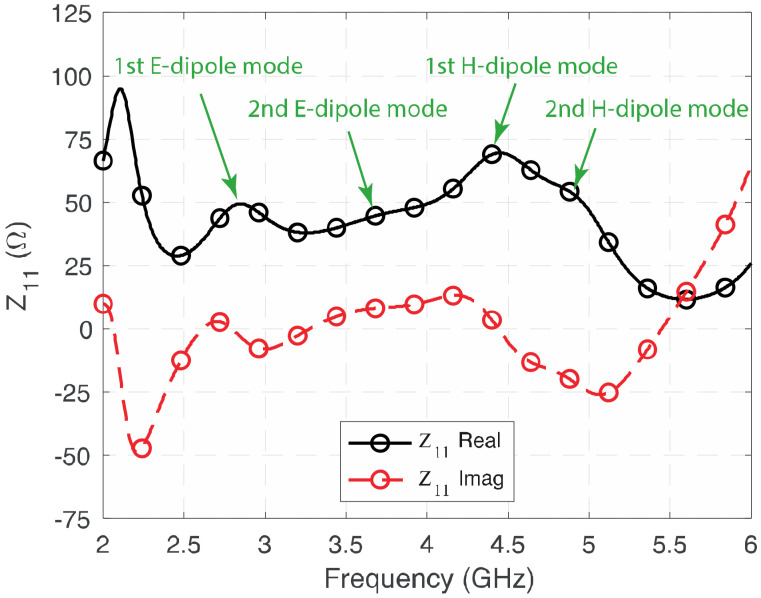
Simulated Z_11_ of the proposed wideband CP cavity-less ME dipole antenna.

**Figure 4 sensors-23-01067-f004:**
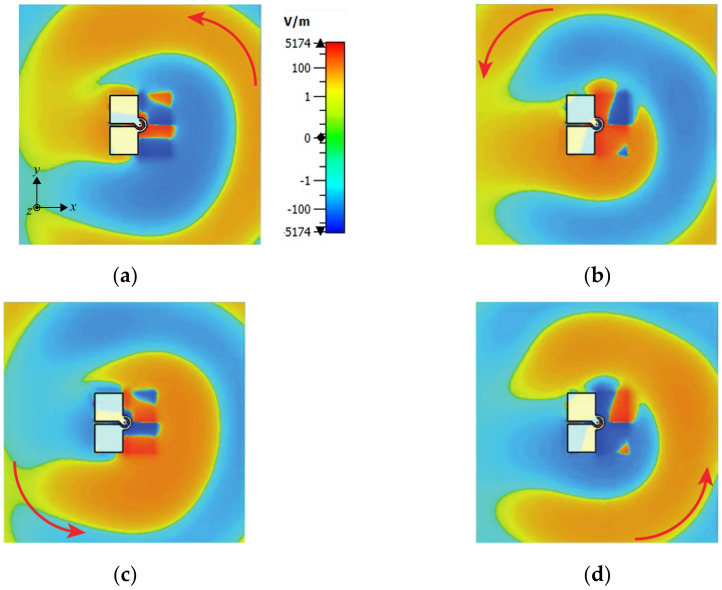
The z-components of E-field of the proposed antenna (**a**) *t* = *T*/8; (**b**) *t* = 3*T*/8; in *xy*-plane; (**c**) *t* = 5*T*/8, (**d**) *t* = 7*T*/8, at 4 GHz.

**Figure 5 sensors-23-01067-f005:**
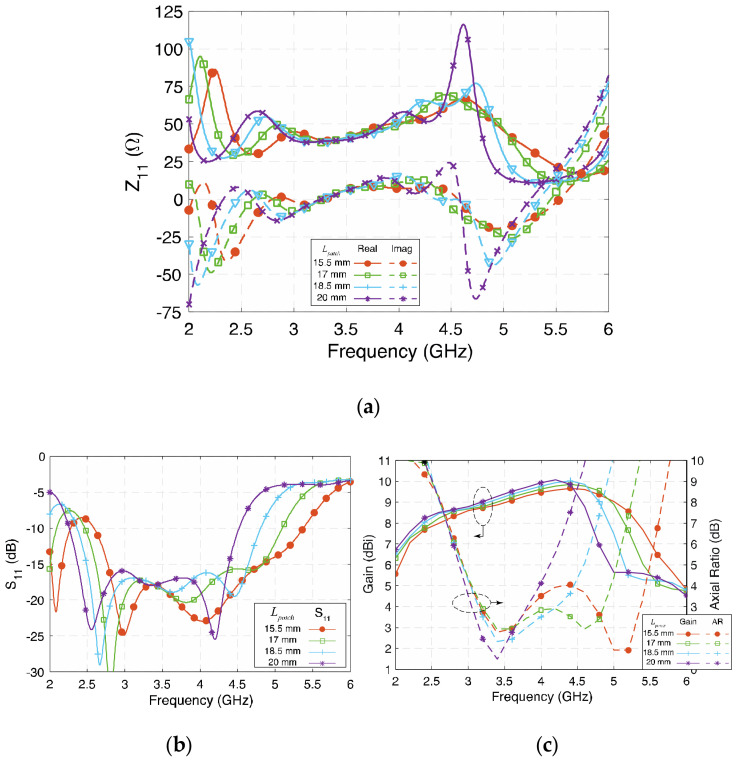
(**a**) Impedance (Z_11_), (**b**) reflection coefficient (S_11_), and (**c**) gain and axial ratio of the proposed antenna at different *L_patch_*.

**Figure 6 sensors-23-01067-f006:**
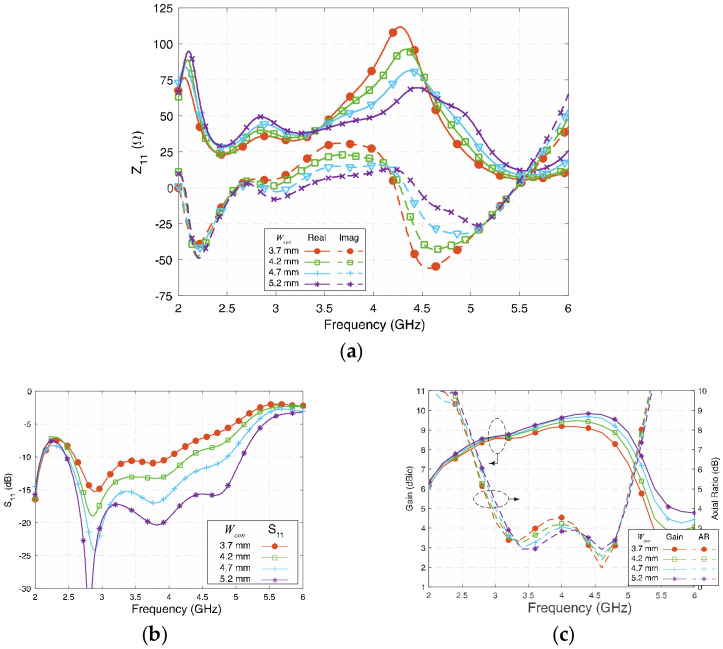
(**a**) Impedance (Z_11_), (**b**) reflection coefficient (S_11_), and (**c**) gain and axial ratio of the proposed antenna at different *W_con_*.

**Figure 7 sensors-23-01067-f007:**
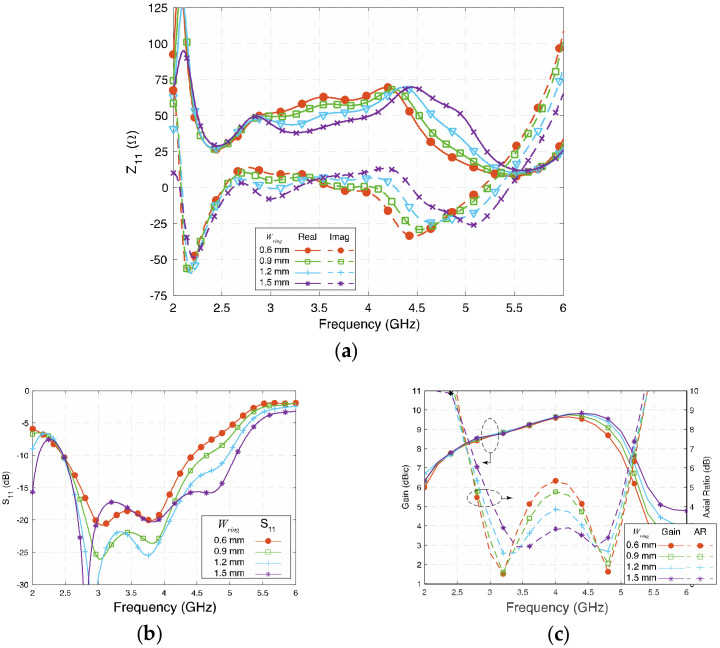
(**a**) Impedance (Z_11_), (**b**) reflection coefficient (S_11_), and (**c**) gain and axial ratio of the proposed antenna at different *W_ring_*.

**Figure 8 sensors-23-01067-f008:**
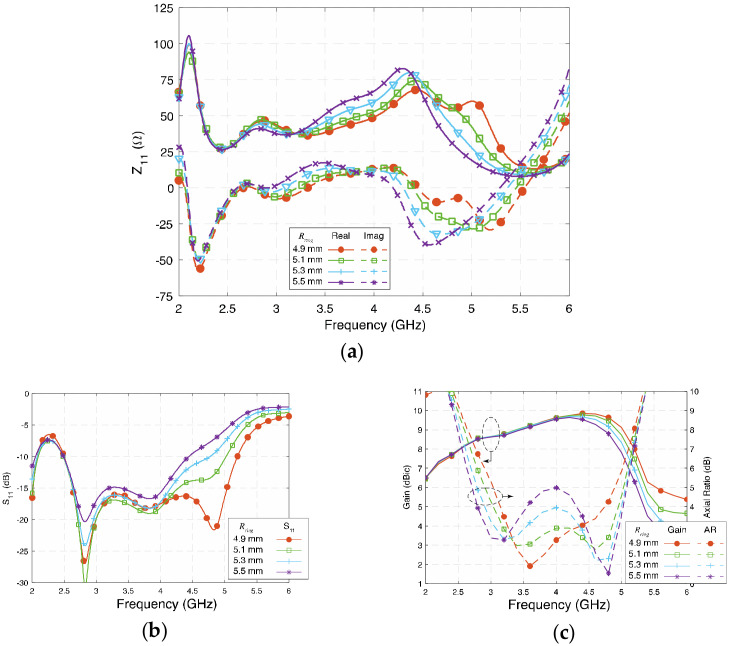
(**a**) Impedance (Z_11_), (**b**) reflection coefficient (S_11_), and (**c**) gain and axial ratio of the proposed antenna at different *R_ring_*.

**Figure 9 sensors-23-01067-f009:**
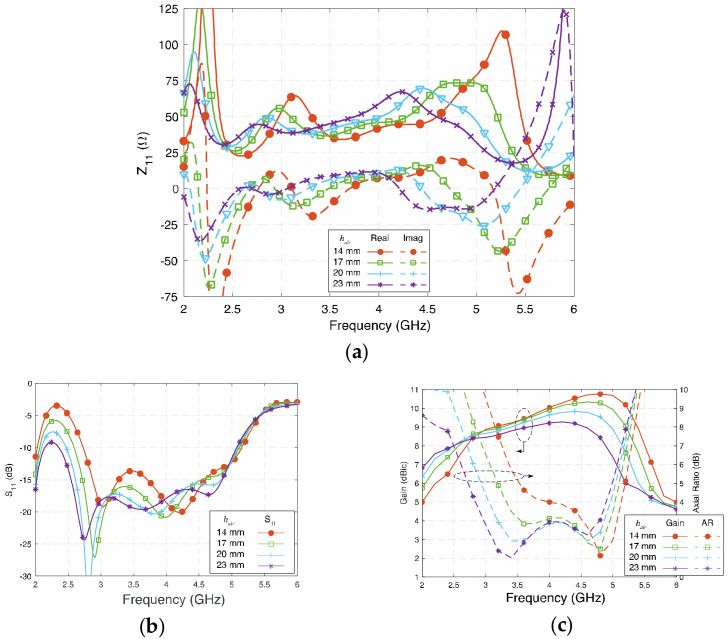
(**a**) Impedance (Z_11_), (**b**) reflection coefficient (S_11_), and (**c**) gain and axial ratio of the proposed antenna at different *h_air_*.

**Figure 10 sensors-23-01067-f010:**
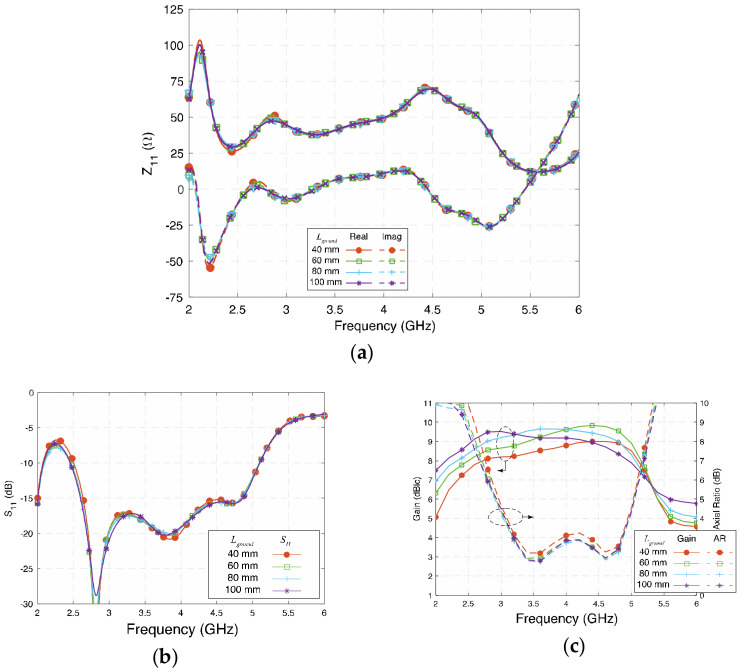
(**a**) Impedance (Z_11_)**,** (**b**) reflection coefficient (S_11_), and (**c**) gain and axial ratio of the proposed antenna at different *L_ground_*.

**Figure 11 sensors-23-01067-f011:**
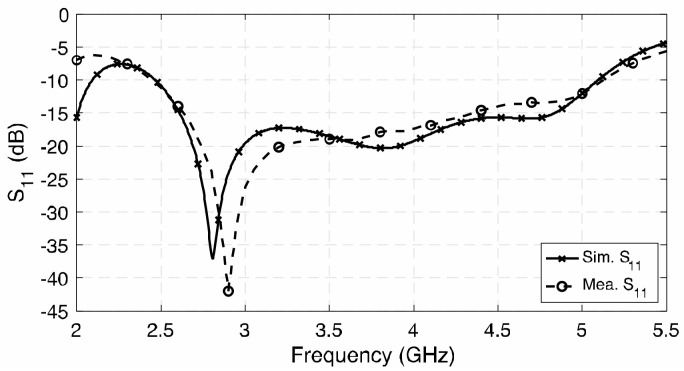
Simulated and measured S_11_ of the proposed antenna.

**Figure 12 sensors-23-01067-f012:**
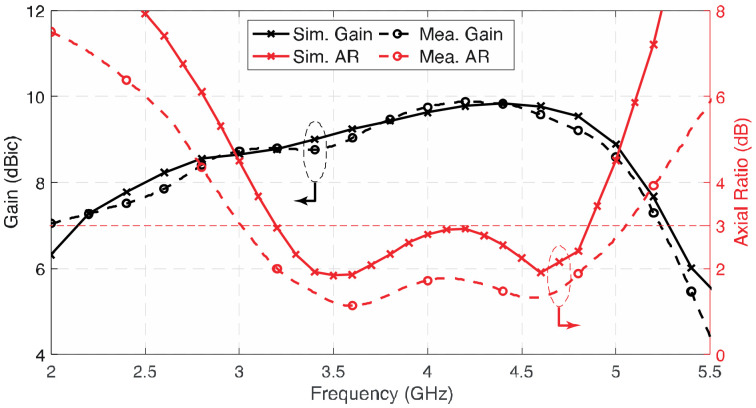
Simulated and measured gain and axial ratio of the proposed antenna.

**Figure 13 sensors-23-01067-f013:**
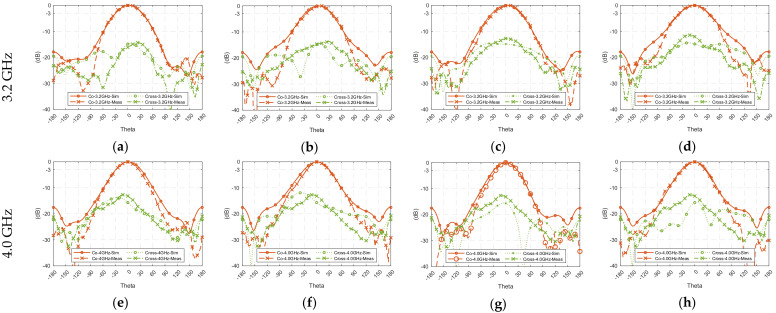
Radiation patterns of the proposed antenna, (**a**–**d**) at 3.2 GHz, (**e**–**h**) at 4 GHz, and (**i**–**l**) at 4.8 GHz, in the four principal planes: *ϕ* = 0°, 45°, 90°, and 135°.

**Table 1 sensors-23-01067-t001:** Comparison of measured results of recently reported wideband circular polarized antennas.

	BW (%)	AR BW (%)	Inline BW (%)	Gain (dBic)	Feed Pt. No.	Footprint (λ_o_^2^)	Height (λ_o_)	Cavity Reflector
[21]	57.6	51.9	51.9	9.50 ± 0.30	2	1.43	0.18	nil
[22]	57.6	38.9	38.9	9.35 ± 1.35	2	0.83	0.26	nil
[23]	59.8	26.8	26.8	8.0 ± 0.50	1	0.41	0.16	single
[24]	57.3	50.9	43.5	9.56 ± 1.19	1	1.02	0.27	single
[25]	69.0	58.6	58.6	8.51 ± 0.89	1	0.63	0.27	single
[26]	93.1	87.6	83.2	3.25 ± 5.38	1	1.24	0.42	nil
[27]	61.8	51.6	51.6	3.70 ± 0.65	1	0.18	0.07	nil
[28]	77.7	68.1	68.1	3.63 ± 4.63	1	0.83	0.22	nil
[29]	79.4	66.7	66.7	8.51 ± 1.29	1	0.74	0.36	complex
[30]	56.7	38.9	38.9	9.35 ± 1.35	2	0.83	0.26	complex
[31]	50	56.5	50	6.7 ± 1.75	1	1.58	0.16	nil
[32]	65.1	71.5	65.1	8.5 ± 1.0	1	1.98	0.26	nil
[33]	22.6	6.8	6.8	10.1 ± 0.1	1	6.67	0.34	nil
This work	70.2	51.5	51.5	9.31 ± 0.56	1	1.16	0.24	nil

**Table 2 sensors-23-01067-t002:** Measured and simulated 3-db beamwidth at low-, mid-, and high-frequency points.

3-dB Beam-Width (°)	3.2 GHz		4.0 GHz		4.8 GHz		|∆_1_| (°)	
	mea.	sim.	mea.	sim.	mea.	sim.	mea.	sim.
*ϕ* = 0°	65.5	59.1	70.3	67.9	74.7	67.3	9.2	8.8
*ϕ* = 45°	63.7	59.9	68.5	65.1	75.8	67.8	12.1	7.9
*ϕ* = 90°	66.0	63.2	69.9	64.3	76.8	69.0	10.8	5.8
*ϕ* = 135°	68.0	63.0	69.0	62.9	77.5	76.3	9.5	13.3
|∆_2_| (°)	4.3	4.1	1.8	5.0	2.8	9.0		

## Data Availability

Please contact Prof. Kin-Fai Tong at k.tong@ucl.ac.uk.

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
