# Peer review of "A Single-Fed Wideband Circularly Polarized Cross-Fed Cavity-Less Magneto-Electric Dipole Antenna"

_sensors, 2023, doi:10.3390/s23031067_

Round 1

Reviewer 1 Report

The article presents a well-known structure, but the results obtained were good in axial ratio and reflection coefficient bandwidth. For this reason, the paper could represent an addition to this research field.

I think it could be better if the author added other references in which magnet-electric dipole was realized with different techniques such as:

- G. Scalise, L. Boccia, G. Amendola, M. Rousstia and A. Shamsafar, "Magneto-Electric Dipole antenna for 5-G applications," 2020 14th European Conference on Antennas and Propagation (EuCAP), 2020, pp. 1-3, doi: 10.23919/EuCAP48036.2020.9136068.

-C. Mustacchio, L. Boccia, R. Maggiora, E. Arnieri and G. Amendola, "E-Band Magneto-Electric Dipole Antenna for 5G Backhauling Applications," 2022 52nd European Microwave Conference (EuMC), 2022, pp. 816-819, doi: 10.23919/EuMC54642.2022.9924392.

Author Response

Many thanks for the reviewer's comments.  The two suggested references have been added to the paper.  We also want to mentioned that although the geometry of the proposed antenna looks similar to some of the antennas reported in the literature, their working principles are different, also per the comparison table provided, the overall impedance bandwidth, gain bandwidth and stability and axial ratio bandwidth and complexity of the proposed antenna are better than those already reported. 

Reviewer 2 Report

Authors present a new circularly polarized antenna suitable for 5G mid-band in the range 2.45-5.1 GHz.  Performance looks quite attractive, but it is not clear for what application it can be used.  The overall size is 80x80x20 mm3 and beampattern about ±30 degrees mean that it can’t be used in the handset.  A large part of manuscript describes numerical modelling of parameters in dependence on dimensions of the elements of antenna.  Optimization is taken for one dimension taking the other from separate modelling.  It would be advantageous if authors present final calculation with optimization by all 6 parameters together, CST can provide such calculation. Usually final 6-parameter optimization can bring additional improvement and explain the difference between simulated and measured performance.  In Fig.13 are presented radiation patterns for 4 fixed principal planes 0, 45, 90, 135 degrees, but for clear understanding of the beampattern it would be advantageous to present a 3-dimention fq view with sidelobes.

Author Response

Many thanks for the reviewer's comments.  As the reviewer mentioned, the dimension and radiation pattern of proposed antenna are more suitable for basestations, rather than handhelds.   Per the operating frequency, the proposed antenna can be used in many sub-6 GHz band wireless systems, in addition to the 5G mobile.  Also, one of the reasons for showing the detailed parametric study is to provide research references to other antenna engineers, so they can design their antennas in other frequency bands for different applications. We believe the impact will be higher than focusing on a specific application.

The dimensions of the final antenna prototypes in the paper has been optimized based on the parametric study.  The parametric study are based on the optimized models for showing the relations between the parameters and antenna performance.

For a single antenna element of size less than one wavelength square, it is unlikely that the antenna has sidelobes,  so we think it is not really necessary to show the 3D radiation pattens in the paper for keeping its clarity.  However, for the reviewer's reference we attached the simulated 3D radiation patterns at the upper, middle and lower frequency points in this response.

Reviewer 3 Report

Although the idea of combining patch and 'slot-like' antennas to realize broadband and dual- or circular polarized antennas is not new, the paper is an interesting contribution in this field. Overall, the paper is well written and good to understand.

Please explain the measurement procedures in more detail. How did you measure the gain (and with which accurracy) and how did you measure the axial ratio? It is not necessarily given that the polarization ellipse is aligned with the main planes.

Some more hints you may find in the uploaded pdf file.

Author Response

Many thanks for the reviewer's comments and precious time reading the paper.  The comments and suggestions have improved the quality of the paper. 

All the comments have been addressed in the revised version, the corrections are highlighted in green in the attached file for the reviewer's reference.

Reviewer 4 Report

The authors presented a circularly polarized cross dipole antenna. The design uses a pair of complementary magneto-dipole modes by utilizing the two open slots formed between the four cross-fed microstrip patches. This method has been widely studied in the literature [R1]. However, the contribution and novelty of this design is limited compared to the existing cross dipole antennas [R3-R5]. Moreover, the performance comparison with state-of-the-art magneto-electric dipole antennas is missing. In conclusion, the presented work does not use novel methods of design not the performance is better compared to similar cross-dipole antennas.

[R1] Y. He, W. He and H. Wong, "A Wideband Circularly Polarized Cross-Dipole Antenna," in IEEE Antennas and Wireless Propagation Letters, vol. 13, pp. 67-70, 2014, doi: 10.1109/LAWP.2013.2296324.

[R2] S. X. Ta and I. Park, “Crossed dipole loaded with magneto-electric dipole for wideband and wide-beam circularly polarized radiation,” IEEE Antennas Wirel Propag Lett, vol. 14, pp. 358–361, 2015.

[R3] H. H. Tran and I. Park, “wideband circularly polarized cavity-backed asymmetric crossed bowtie dipole antenna,” IEEE Antennas Wirel Propag Lett, vol. 15, pp. 358–361, 2016.

[R4] H. H. Tran, I. Park, and T. K. Nguyen, “Circularly polarized bandwidth-enhanced crossed dipole antenna with a simple single parasitic element,” IEEE Antennas Wirel Propag Lett, vol. 16, pp. 1776–1779, 2017.

[R5] T. K. Nguyen, H. H. Tran, and N. Nguyen-Trong, “A wideband dual-cavity-backed circularly polarized crossed dipole antenna,” IEEE Antennas Wirel Propag Lett, vol. 16, pp. 3135–3138, Oct. 2017.

Author Response

Many thanks for the reviewer's comments.  We also want to clarify that although the geometry of the proposed antenna looks very similar to the antenna reported in [R1], their working principles are different. The title of [R1] is "A wideband circularly polarized cross-dipole antenna"; it operates purely as a cross-dipole antenna, rather than a ME-dipole antenna.  As the slots are not aligned, the potential M-dipole modes have not been utilised, more importantly, the 3dB AR bandwidth hasn't been enhanced as well. As shown in the state-of-the-art comparison Table 1, the overall impedance bandwidth, inline bandwidth, gain stability and axial ratio bandwidth of the proposed antenna are better than that in [R1].

For [R2-R5], they do have their individual contributions and novelty in the wide 3dB AR bandwidth antennas, however, they suffer from either lower gain, narrower impedance bandwidth, narrower AR bandwidth [R2], or larger gain variation in the passband [R3-R5] when compared to the proposed antenna, more importantly they all require a cavity reflector to support their performance.  On the contrary, the proposed extra E-dipole mode and M-dipole created by the proposed geometry provide a more comprehensive performance which extend the impedance bandwidth and axial ratio bandwidth without sacrificing the stable high gain, this technique has not been reported in the literature; the method is novel. It is not fair to say the contribution and novelty of the proposed antenna is limited when compared to [R2-R5]. 

The comparisons to state-of-the-art circularly polarised magneto-electric dipole antennas have been added.

Round 2

Reviewer 2 Report

Authors improved the manuscript according to previous comments and recommendations and this vrsion can be published in present form

Reviewer 4 Report

Although the authors have revised the manuscript, however, my concerns regarding the contribution and novelty of this work still persist. The novelty is marginal. 

The reviewer believes that this work is the optimization of the [R1](optimization of the gap between the two cross dipoles). The antenna geometry is almost the same. 

[R1] Y. He, W. He, and H. Wong, "A Wideband Circularly Polarized Cross-Dipole Antenna," in IEEE Antennas and Wireless Propagation Letters, vol. 13, pp. 67-70, 2014, doi: 10.1109/LAWP.2013.2296324.
